# Coherence-controlled chaotic soliton bunch

Ze-Xian Zhang[1,2], Min Luo[1,2], Jia-Hao Liu[1,2], Yi-Tao Yang[1,2], Ti-Jian Li[1,2], Meng Liu[1,2], Ai-Ping Luo[1,2], Wen-Cheng Xu[1,2] & Zhi-Chao Luo [1,2] ✉

Controlling the coherence of chaotic soliton bunch holds the promise to explore novel light-matter interactions and manipulate dynamic events such as rogue waves. However, the coherence control of chaotic soliton bunch remains challenging, as there is a lack of dynamic equilibrium mechanism for stochastic soliton interactions. Here, we develop a strategy to effectively control the coherence of chaotic soliton bunch in a laser. We show that by introducing a lumped fourth-order-dispersion (FOD), the soliton oscillating tails can be formed and generate the potential barriers among the chaotic solitons. The repulsive force between neighboring solitons enabled by the potential barriers gives rise to an alleviation of the soliton fusion/annihilation from stochastic interactions, endowing the capability to control the coherence in chaotic soliton bunch. We envision that this result provides a promising test-bed for a variety of dynamical complexity science and brings new insights into the nonlinear behavior of chaotic laser sources.

Optical solitons are self-localized wave structures arising from a balance between dispersion and nonlinearity[1,2], which lays the foundations for the development of a variety of applications, including high-energy pulsed lasers[3,4], optical frequency comb[5], and supercontinuum generation[6]. Therefore, understanding the fundamental physics hidden behind solitons is critically important. Albeit propagating like waves, optical solitons can interact in a particle-like fashion[7,8]. Depending on the initial phases and profiles, plenty of soliton interactions have been reported and observed, manifesting rich behaviors such as fusion, fission, and annihilation[9–11]. The interacting solitons may also give rise to self-localized structures, namely multi-soliton complexes[12]. Through strongly short- or long-range interactions, the multi-soliton complexes can even evolve into a chaotic yet localized soliton bunch[13–15].

Recent research in the localized chaotic soliton bunch indicates that it can be regarded as a promising physical platform for investigating the dynamics of rogue waves and turbulence[16–18]. Aside from the fundamental research, the properties of high energy and partially coherent broadband spectrum enable the chaotic soliton bunch to find practical applications such as speckle-free imaging[19] and supercontinuum generation[20]. For applications, controlling the coherence of chaotic soliton bunch is highly demanded, as the coherence acts as a

critical variable that influences the performance of photonic systems, for example, manipulation of rogue wave generation[21] and chaotic LiDAR[22,23]. In retrospect, the chaotic soliton bunch related experiments were mainly performed in nonlinear optical systems dominated by group-velocity dispersion (GVD)[13–18]; that is, the pulse shaping relies on the balance between Kerr nonlinearity and GVD. In fact, the chaotic solitons formed in these systems exhibit randomly evolving phases and amplitudes, causing numerous and stochastic fusions/annihilations during the collision process. Owing to a lack of a dynamic equilibrium mechanism for stochastic soliton interactions within the bunch, it remains a challenging task to control the coherence of a chaotic soliton bunch[24].

The key to controlling the bunch-to-bunch phase coherence of chaotic solitons is to build an equilibrium force in the process of soliton collisions that can be used to adjust the degree of fusions/annihilations. In fact, optical solitons can also be stabilized in the presence of fourth-order dispersion (FOD)[25–27], and even by a balance of pure negative-FOD and positive Kerr nonlinearity. This study leads to a new concept of pure-quartic soliton (PQS)[28,29]. Due to an imperfect counterbalance between self-phase modulation (SPM)- and FOD-induced phase shifts, the oscillating tails appear on both edges of the soliton[28,29]. From a general perspective of soliton interactions, the

[1]Guangdong Provincial Key Laboratory of Nanophotonic Functional Materials and Devices, Guangdong Basic Research Center of Excellence for Structure and Fundamental Interactions of Matter, School of Information and Optoelectronic Science and Engineering, South China Normal University, Guangzhou, Guangdong 510006, China. [2]Guangzhou Key Laboratory for Special Fiber Photonic Devices and Applications, South China Normal University, Guangzhou, Guangdong 510006, China. ✉e-mail: zcluo@scnu.edu.cn

oscillating tails of FOD-driven soliton would play a vitally important role in soliton interactions[30]. As the oscillating tails can create a potential barrier, a repulsive force between adjacent solitons can be effectively generated, thereby mitigating the strong soliton fusion/annihilation during the collision process. Therefore, inspired by the FOD engineering on the soliton shaping, a fundamental question naturally arises as to whether new scenarios can be discovered for coherence control on chaotic soliton bunch.

In the present work, we directly address this question by revealing the dynamics of chaotic soliton bunch in a fiber laser with lumped FOD engineering through a combined experimental and numerical study. The oscillating tails of FOD-driven soliton effectively create a potential barrier that generates a repulsive force between neighboring solitons. This property enables the chaotic soliton bunch to reduce collision-induced fusion/annihilation, leading to a partial preservation of bunch-to-bunch phase coherence despite the chaotic motion of solitons. In

particular, by manipulating the intensities of oscillating tails, we found that the coherence of chaotic soliton bunch can even be flexibly controlled. These results represent a crucial step towards developing coherence-controlled chaotic optical systems as well as new types of chaotic laser sources.

## Results
### Operation principle of coherence-controlled chaotic soliton bunch
The optical soliton with oscillating tails can be generated in nonlinear optical systems by incorporating an FOD component (Fig. 1a, also see Supplementary Note 1 for the formation mechanism of oscillating tails). As a nonlinear optical system for studying the dynamics of chaotic soliton bunch, a fiber laser passively mode locked by a commercial saturable absorber (SA, Batop, SA-1550-35-2ps) is adopted. For implementation, the introduction of FOD and the compensation of

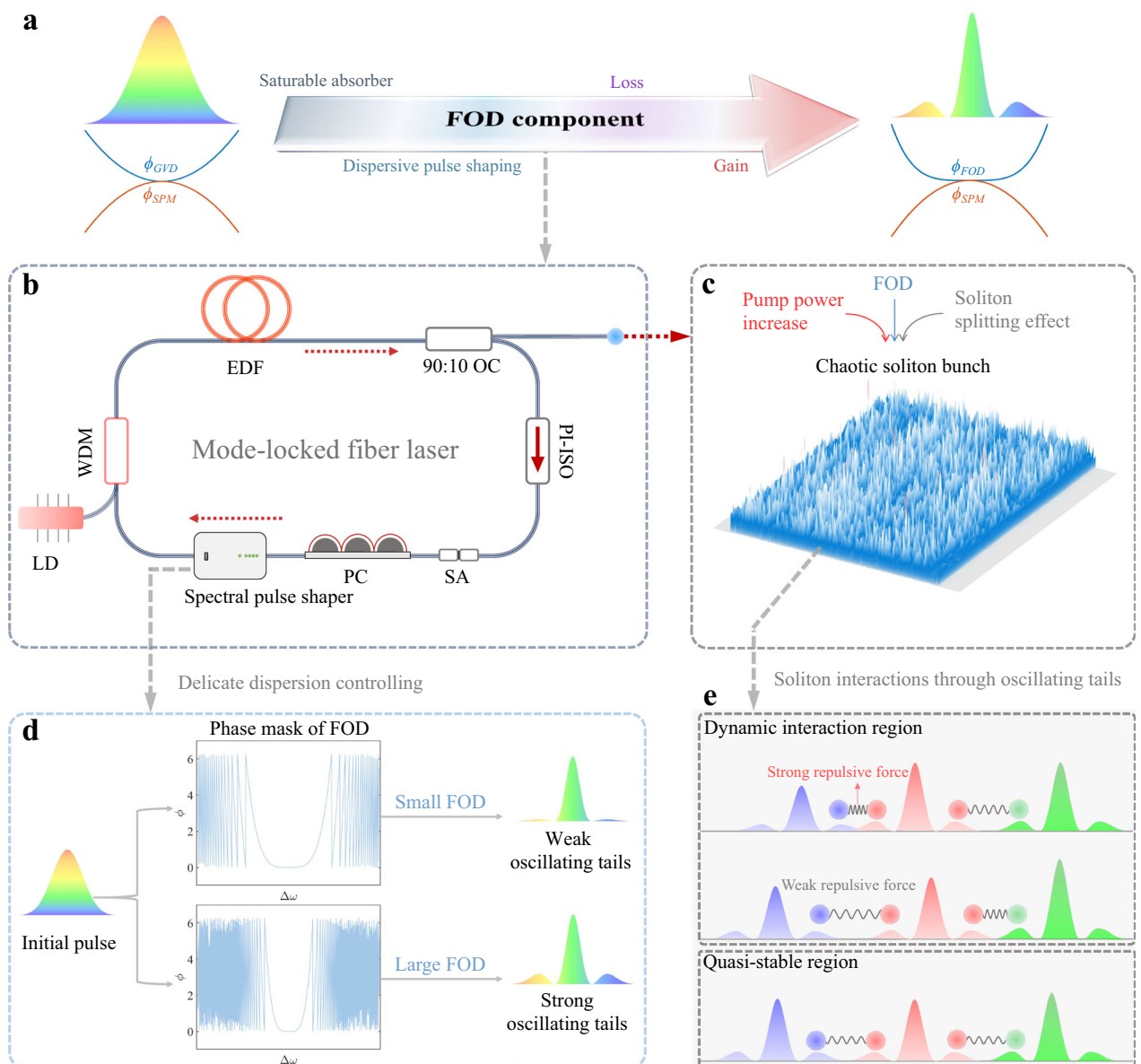

**Fig. 1 | Principle of the coherence-controlled chaotic soliton bunch in a laser. a** Concept of soliton generation with oscillating tails. **b** Experimental setup of mode-locked fiber laser with the following components: spectral pulse shaper; LD, 976 nm laser diode; WDM, wavelength division multiplexer; EDF, erbium-doped fiber; OC, optical coupler; PI-ISO, polarization insensitive isolator; SA, saturable

absorber; PC, polarization controller. **c** Sketch of chaotic soliton bunch. **d** Principle of the control of oscillating tails using a spectral pulse shaper. **e** Conceptual illustration of the interaction scenarios between adjacent solitons through repulsive force created by oscillating tails.

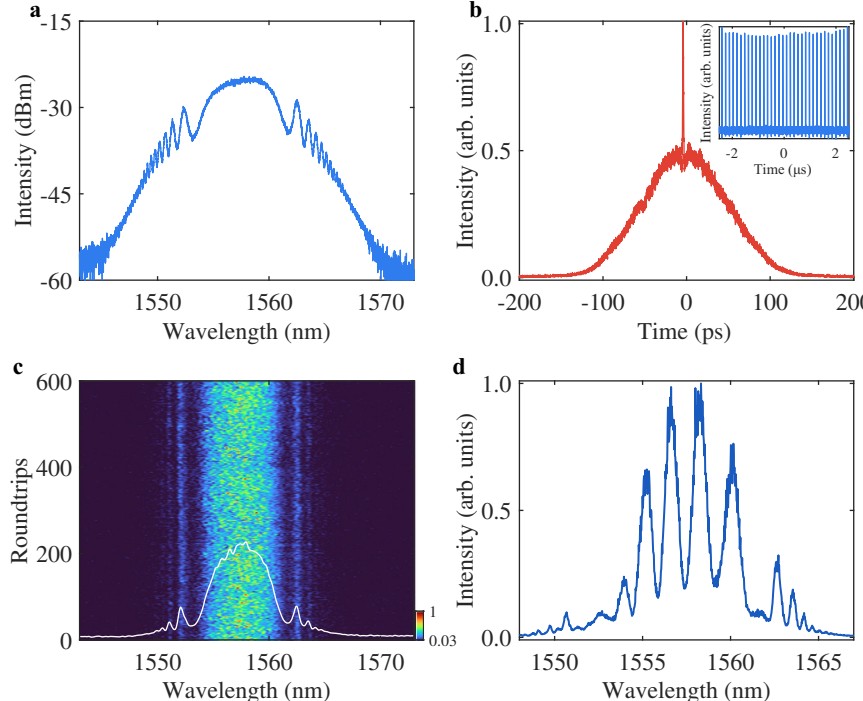

**Fig. 2 | Experimental results of FOD-driven chaotic soliton bunch in the fiber laser. a** Averaged spectrum recorded with an OSA. **b** Autocorrelation (AC) trace, inset: pulse train. **c** Experimentally measured shot-to-shot spectra of 600 consecutive roundtrips with the DFT technique. Inset with white curve: spectrum obtained by averaging the 600 roundtrips of shot-to-shot spectra. **d** Spectral interference pattern of chaotic soliton bunches, using M-Z interferometer.

GVD are realized by a spectral pulse shaper (Finisar, Waveshaper 4000 A), as illustrated in Fig. 1b (see 'Methods' section for details). Under proper cavity parameters, the fiber laser delivers a chaotic soliton bunch owing to the overdriven nonlinear effect[14] (Fig. 1c). Figure 1e conceptually illustrates the scenarios of soliton interactions, where we can see that the potential barrier created by oscillating tails generates a strong repulsive force when they approach each other closely. This ability enables the solitons to prevent wave breaking to some extent despite chaotic motion. Moreover, imposing different phase masks of FOD can control the degree of imperfect counterbalance between SPM- and FOD-induced phase shifts, leading to an adaptive energy flow between oscillating tails and the main lobe of optical soliton[31]. In this case, the intensities of oscillating tails can be tailored (Fig. 1d, also see Supplementary Note 2 for details). Note that the magnitude of the force between neighboring solitons is determined by the intensities of oscillating tails and their overlap degree. Therefore, we are able to manipulate the soliton interactions that facilitate the coherence-controlled operation of chaotic soliton bunch.

**Experiments with coherence control of a chaotic soliton bunch**
To enable the fiber laser to deliver solitons with oscillating tails, initially the FOD is set to $\beta_4 = -20$ ps$^4$ km$^{-1}$ and the GVD is fully compensated by virtue of the spectral pulse shaper. The fiber laser can step into either passive mode locking with single soliton operation or localized chaotic soliton bunch regime, depending on the settings of cavity parameters such as pump power level. Then, the pump power was set to 130 mW, and we focused on the dynamics of the chaotic soliton bunch regime. Figure 2 summarizes the spectral and temporal performance of the chaotic soliton bunch. Compared to the smooth spectrum of the stable mode-locking, the spectrum of the chaotic soliton bunch shows noisy spikes across the whole profile (Fig. 2a). However, despite the chaotic behavior of the soliton bunch, a notable phenomenon is that the Kelly sidebands resulted from the interference

between soliton and dispersive waves[32] can still be observed on the averaged spectrum, which is distinguished from chaotic pulse bunch supported by GVD-dominated fiber laser (see Supplementary Note 3). The appearance of Kelly sidebands originated from the incomplete destruction of the interference between soliton and dispersive waves, indicating that the coherence of the chaotic soliton bunch is partially preserved. To further verify the operation regime of the fiber laser, we also provide the measured autocorrelation trace (Fig. 2b). The large pedestal of the autocorrelation trace suggests that the fiber laser indeed operates in the chaotic soliton bunch regime[13]. The pulse train is also plotted (inset of Fig. 2b), which shows an evident intensity fluctuation owing to the partially coherent property.

As the chaotic soliton bunch evolves dramatically with time and the optical spectrum analyzer (OSA) exposes the average spectral measurement, the dispersive Fourier transformation (DFT) technique[33,34] is applied to reveal the real-time spectral dynamics of the chaotic soliton bunch against cavity roundtrips (see 'Methods' section). The real-time spectral evolution over 600 roundtrips is unwrapped in a spatiotemporal manner (Fig. 2c), which exhibits the shot-to-shot instability on the spectral profiles. Although the Kelly sidebands disappear after certain roundtrips, they still can be traced on both sides of the shot-to-shot spectra over a sustainable observation time. These results clearly demonstrated the sustainable interference between soliton and dispersive waves despite the chaotic soliton interactions within the bunch, which can be attributed to the fact that the oscillating tails of mode-locked solitons avoid strong collisions among them. In fact, the phase coherence of the chaotic soliton bunch across the whole spectrum can be quantified by using a Mach–Zehnder interferometer[35] (see 'Methods' section). In the past, such a chaotic soliton bunch supported by GVD generally exhibited an absence of phase correlation and, thereby, no spectral interference pattern at the output of the interferometer[24]. However, evident interference fringes can be observed on the spectrum of FOD-driven chaotic soliton bunch (Fig. 2d), demonstrating the ability of coherence

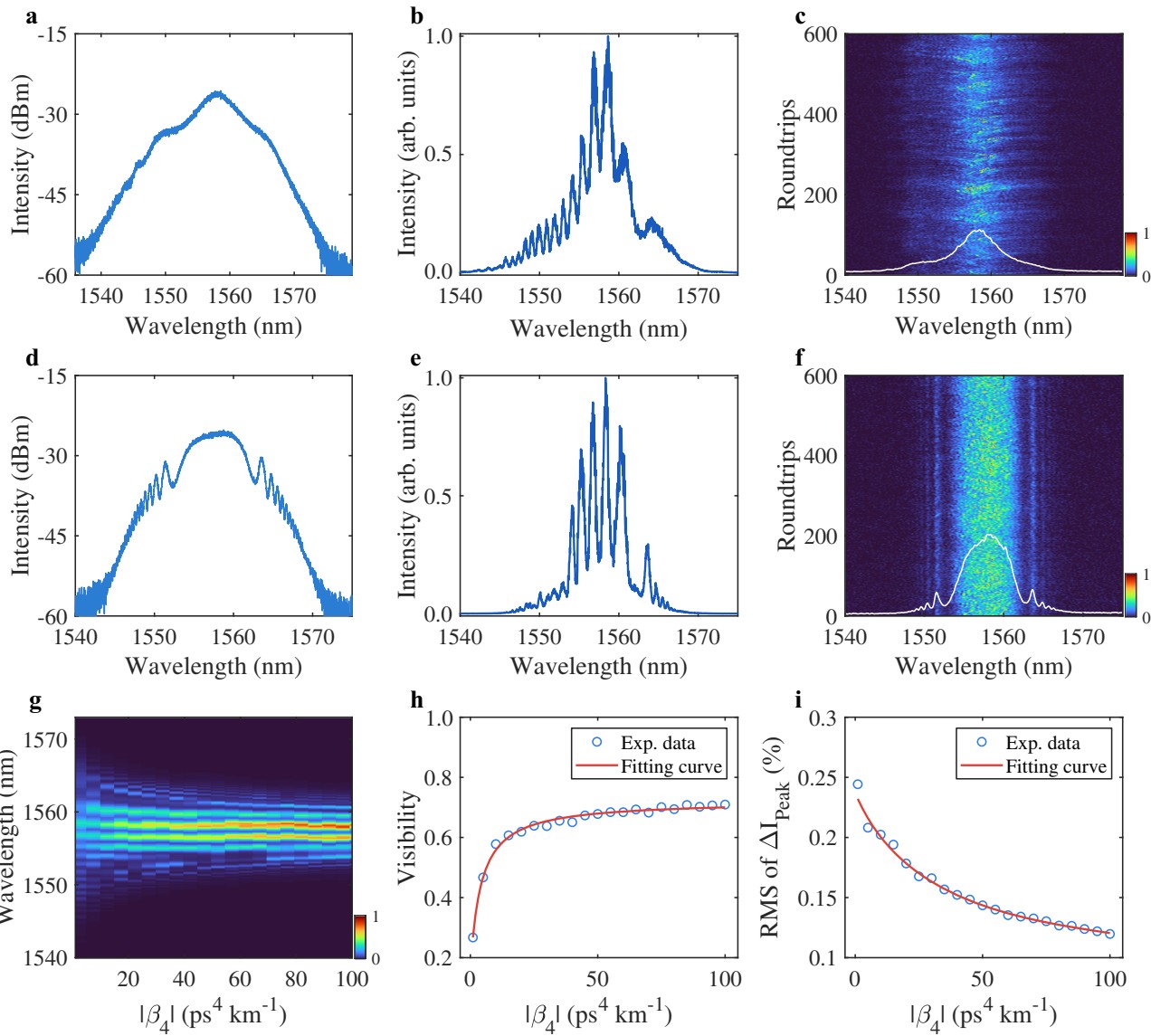

**Fig. 3 | Experimental results of coherence-controlled chaotic soliton bunch.**
**a**–**f** Characteristics of chaotic soliton bunch with $\beta_4 = -1\,\mathrm{ps^4\,km^{-1}}$ (**a**–**c**) and $\beta_4 = -10\,\mathrm{ps^4\,km^{-1}}$ (**d**–**f**), respectively. **a**, **d** Averaged spectra. **b**, **e** Spectral interference patterns. **c**, **f** Shot-to-shot spectra over 600 roundtrips, inset with white curve: the averaged spectra computed from the single-shot measurements. Evolution of **g** spectral interference pattern and **h** visibility and **i** RMS of spectral peak variation ($\Delta I_{\mathrm{peak}}$) with an increasing $|\beta_4|$.

preservation. Note that the fringe visibility has reached 0.62, suggesting that the bunch-to-bunch coherence is partially preserved.

Cavity dispersion plays an important role in the formation and evolution of soliton in fiber lasers[36]. In particular, the FOD amount for shaping the soliton is directly linked to the intensities of oscillating tails, which influences the degree of the soliton fusion/annihilation inside the chaotic bunch as well as the coherence property. We then varied the FOD amount ($\beta_4 = -1\,\mathrm{ps^4\,km^{-1}}$ and $\beta_4 = -10\,\mathrm{ps^4\,km^{-1}}$) and captured the corresponding spectral characteristics (Fig. 3a, d), while maintaining all other cavity parameters, i.e., pump power and PC orientation unchanged. From the noisy spikes on the spectrum, it can be inferred that the fiber laser is still operating in the regime of chaotic soliton bunch. Note that the spectral separation between the center wavelength and first-order sidebands is dependent on the net cavity FOD[29]. Therefore, the Kelly sidebands are absent when the FOD is relatively small (Fig. 3a). The corresponding autocorrelation traces, which show the evident pedestals, also verify the operation regime of the chaotic soliton bunch (see Supplementary Note 4 for details).

The corresponding real-time spectral evolutions of chaotic soliton bunches with different FOD amounts are also recorded over 600 consecutive roundtrips (Fig. 3c, f). It further demonstrates that the dramatic shot-to-shot variation does not thoroughly break the interference between the soliton and dispersive waves with a larger FOD (Fig. 3f), illustrating the expected feature of Kelly sidebands. The phase coherences dominated by two FOD values are also characterized by interfering with the neighboring chaotic soliton bunches (Fig. 3b, e), where the fringe visibility is evidently enhanced with a larger FOD. To further reveal the coherence property of chaotic soliton bunch, Fig. 3g, h present the bunch-to-bunch phase coherence with the varying FOD from $-1\,\mathrm{ps^4\,km^{-1}}$ to $-100\,\mathrm{ps^4\,km^{-1}}$. The fringe visibility of the chaotic soliton bunches with different FODs ranges from 0.27 to 0.71. Here, we can see that the fringe visibility does not linearly change with the increasing FOD. It is because the effect of FOD on pulse shaping can be qualitatively reflected by the dispersion length, which is defined as $L_{\mathrm{FOD}} = T_0^4/|\beta_4|$. This leads to a nonlinear relationship between the fringe visibility and the FOD amount. Moreover, although a lumped

FOD engineering is adopted, the calculated dispersion length (see Supplementary Note 5) indicates that the effect of FOD on pulse shaping is still evident during the process of coherence-controlled operation when compared to the laser cavity length. We note that an increasing FOD leads to a larger intensity of oscillating tails of soliton. In turn, the oscillating tails with a larger intensity result in a higher potential barrier and a stronger repulsive force, which endows the chaotic soliton bunch with a stronger ability to alleviate soliton fusion/annihilation during the collision process. In this scenario, the degree of bunch-to-bunch phase coherence can be manipulated with a varying FOD.

Meanwhile, we recorded the highest spectral peak of the DFT spectrum over 10000 roundtrips and calculated the RMS of spectral peak fluctuations. In fact, the stronger soliton fusion or annihilation will lead to a larger fluctuation of spectral intensity and a deterioration of phase coherence. Consequently, the lower RMS of the spectral peak fluctuation corresponds to a higher bunch-to-bunch phase coherence (Fig. 3i). Moreover, we have also checked the coherence property of chaotic soliton bunch with a certain amount of GVD and a varying FOD (see Supplementary Note 6). Although the residual distributed GVD along the optical fibers will weaken the role of FOD on the pulse shaping, the coherence-controlled operation is still well established. These results clearly verify our hypothesis that the oscillating tails of FOD-driven solitons prevent strong interactions, leading to a partial

preservation of coherence as well as the coherence-controlled property for the chaotic soliton bunch.

## Numerical simulation of chaotic soliton bunch dominated by FOD

As the temporal evolution of the chaotic soliton bunch in the fiber laser is too fast to be observed with state-of-the-art measurement equipment, the numerical simulation is further carried out to uncover the roundtrip-to-roundtrip temporal dynamics. The numerical simulation model is based on the nonlinear Schrödinger equation, which is solved with a standard symmetric split-step Fourier method. The numerical model includes physical terms such as SPM, GVD, FOD, gain coefficient and saturated gain with a finite bandwidth of the EDF, which are set according to our experimental setup (see 'Methods' section). We begin our simulations by setting the pump power to $E_s = 600$ pJ and $\beta_4 = -20$ ps$^4$ km$^{-1}$. The seed signal is a sech-shaped pulse with relatively weak intensity. Owing to an overdriven nonlinear effect, the mode-locked soliton in the laser cavity will split into many pulses and evolve into a chaotic bunch regime. Note that the stable single-soliton operation can be obtained, which is shown in Supplementary Note 7.

Figure 4 illustrates the spectral and temporal dynamics of FOD-driven chaotic soliton bunch in the fiber laser based on our simulation model. The shot-to-shot spectra over 3000 roundtrips present substantial intensity fluctuations (Fig. 4a). Nevertheless, the Kelly

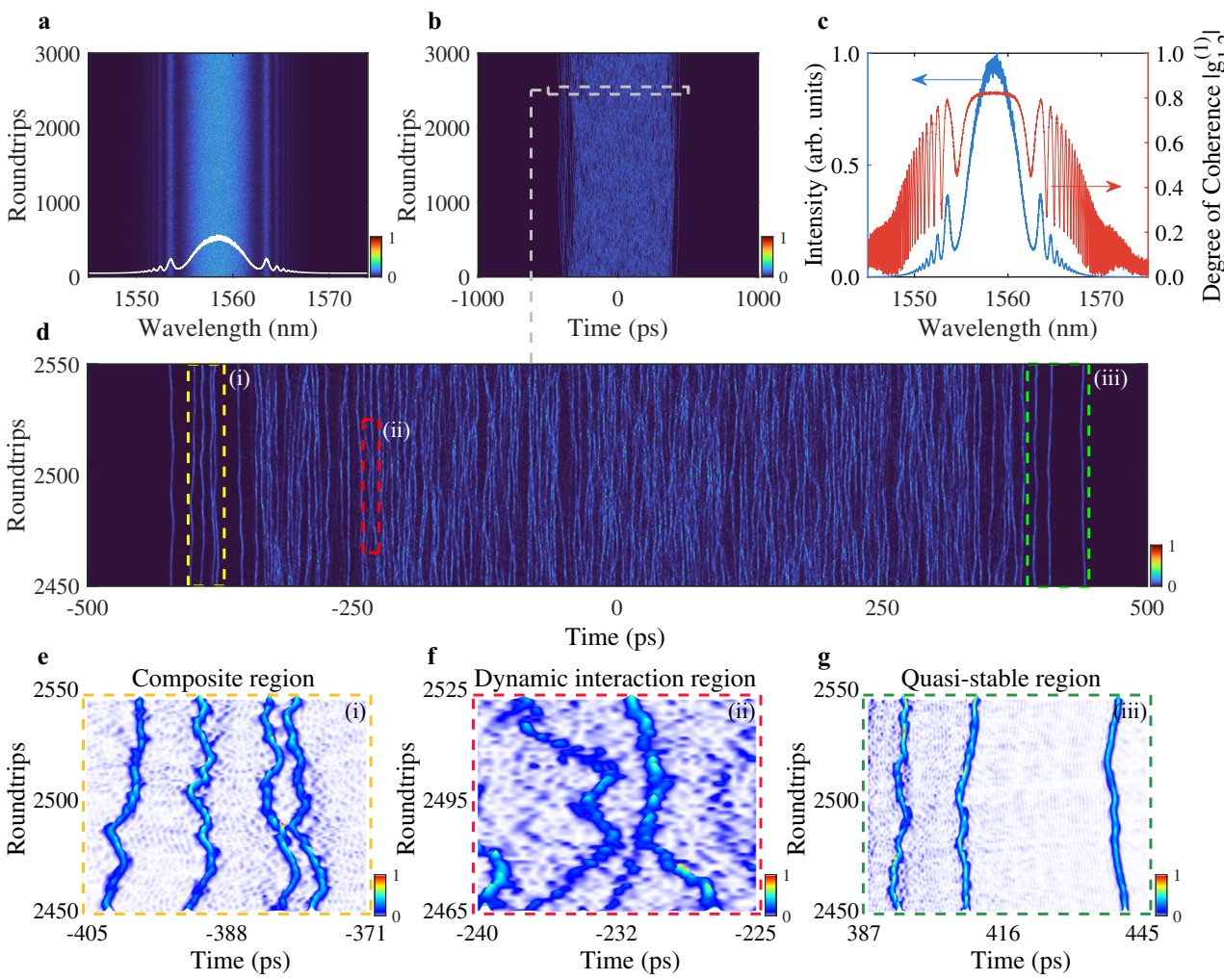

**Fig. 4 | Simulation results of chaotic soliton bunch. a, b** Spectral and temporal evolution of chaotic soliton bunch with $\beta_4 = -20$ ps$^4$ km$^{-1}$. **c** Averaged spectrum (blue curve) and MDOC (red curve). **d** Magnification of white dashed box in (**b**).

Close-ups in different regions, as indicated by the yellow (i), red (ii) and green (iii) dashed boxes in (**d**): **e** Composite region. **f** Dynamic interaction region. **g** Quasi-stable region.

sidebands can still be observed if we trace the spatiotemporal dynamics, which is well consistent with the experimental results. The averaged spectrum over 3000 roundtrips also clearly verifies the appearance of Kelly sidebands, as indicated in the white curve of Fig. 4a. Figure 4b unveils the corresponding temporal evolution, where the numerous solitons are localized well in ~1 ns timescale. In particular, although the pulses localized in the bunch are unstable and suffer from intensity fluctuations, most of the solitons are still able to maintain their propagation along the laser cavity. That is, the number of the solitons inside the bunch does not change much despite the chaotic collisions among them. It can be observed that a repulsive force will be acting on two solitons from a closer distance that avoids the collision-induced fusion/annihilation of pulses, mainly benefitting from the overlap between the oscillating tails in this condition (Fig. 4d).

Figure 4e–g also provides the details of representative soliton interaction scenarios. In addition, theoretical results indicate that the energy can be partially stored in the oscillating tails of FOD-shaping solitons with the increasing pulse peak power, as a result of energy exchange between the main lobe and oscillating tails[31]. Therefore, this feature leads to a higher wave-breaking threshold of solitons, which also contributes to the coherence maintenance of chaotic soliton bunch. Note that the phase coherence can be partially manifested as energy fluctuation. For clarity, we calculate the energy fluctuation of localized soliton bunch (defined as $\Delta E = |E_n - E_{n-1}|/E_{n-1}$) as they propagate in the laser cavity over 7000 roundtrips. The RMS of the bunch-to-bunch energy fluctuation is 0.1335%, which is smaller than that of the chaotic pulse bunch (0.3436%) without the role of FOD. Additional details of energy fluctuation, as well as the spectral and temporal dynamics for the conventional chaotic soliton bunch dominated by GVD are presented in Supplementary Note 8. This result, in turn, further demonstrates that the ability of coherence maintenance for chaotic soliton bunch dominated by FOD is different from that of the GVD-based fiber laser. Moreover, the phase stability of the chaotic soliton bunch across the whole spectrum is quantified by calculating the modulus of the complex degree of first-order coherence (MDOC)[6,35]. The MDOC is calculated to be 0.76 in the center part of the mode-locked spectrum, which reflects the partial coherence property of the FOD-shaping chaotic soliton bunch (Fig. 4c).

Although the results presented in Fig. 4 are associated with a single FOD value, the coherence preservation of chaotic soliton bunch would be a more general phenomenon according to the experimental observations. Moreover, the experimental results indicate that the coherence can be manipulated by imposing different FOD on the chaotic solitons. Therefore, we further simulate the dynamics of chaotic soliton bunch in the fiber laser with different FOD values (Fig. 5). As expected, the Kelly sidebands can be observed if the amount of FOD is large enough (Fig. 5d). Meanwhile, the generated chaotic solitons are well localized in the bunch (Fig. 5b, e). From Fig. 5c, f, the MDOC is higher across the center part of the mode-locked spectrum with a larger FOD. For better clarity, the averaged spectral interference patterns between the neighboring bunches are plotted with the FOD from $\beta_4 = -1.9 \text{ ps}^4 \text{ km}^{-1}$ to $\beta_4 = -100 \text{ ps}^4 \text{ km}^{-1}$ (Fig. 5g). The evolution trend of the interference patterns is in good accordance with the experimental results, where the fringe visibility ranges from 0.18 to 0.92 with the varying FOD (Fig. 5h). Moreover, the RMS of the energy fluctuations corresponding to a varying FOD is also plotted (Fig. 5i). Note that the variation range of the fringe visibility as well as the RMS of the energy fluctuations in the simulations are larger than those of experimental results, which can be attributed to the fact of detection sensitivity and noise in the experiments. Nevertheless, the trends of these two features with the varying FOD are highly consistent with the experimental results.

## Discussion

Although the chaotic soliton bunch, also known as noise-like pulse, has been discovered in fiber lasers for over 25 years[13], the coherence of chaotic soliton bunch is generally believed to be absent during its evolution in the laser cavity. However, our studies provide clear experimental and numerical evidence that the coherence of chaotic soliton bunch dominated by FOD can be partially preserved and flexibly manipulated. Moreover, the basic features of the FOD-driven chaotic soliton bunch appear to be universal and can be observed over a broad range of cavity parameters. In this work, the dynamics of chaotic soliton bunch is limited in the context of fiber laser operating in the regime of negative FOD. Nevertheless, it is expected the positive-FOD-dominated fiber lasers[37,38] will also be a good test-bed for exploring the chaotic interactions of dissipative solitons.

Taking advantage of the profile of FOD-shaping soliton, namely oscillating tails, the soliton interactions inside the bunch, i.e., fusion or annihilation induced by the collisions, can be effectively alleviated owing to the generated potential barrier (repulsive force created by the oscillating tails). It has been suggested that the soliton interaction forces during the collision process, for example, repulsive and attractive forces are also linked to the initial soliton phases and their temporal tails[39], where the phases are somewhat chaotic for the solitons in the bunch. In this way, the collision-induced soliton annihilations cannot be avoided completely; that is, they can only be partially prevented. It is worth noting that the strength of the potential barrier that prevents the soliton interactions (i.e., collisions) is dependent on the intensities of oscillating tails, whereby the intensities of oscillating tails can be adjusted by setting different amounts of FOD. Therefore, the coherence of chaotic soliton bunch is able to be controlled with the adjustment of FOD that is introduced into the fiber laser.

An intuitive application of such a chaotic soliton bunch with the coherence-controlled property is to act as a nonlinear optical system for the manipulation of rogue wave generation, since the coherence is directly linked to the energy fluctuation inside the bunch induced by the stochastic soliton collisions. Utilizing the coherence-controlled chaotic soliton bunch, we have demonstrated the control of the dynamics of rogue waves. Additional experimental details regarding the manipulation of rogue waves are presented in Supplementary Note 9. In fact, a similar concept of coherence-controlled chaotic soliton bunch can also be extended to other chaotic systems, such as complex nonlinear networks[40], chaotic optical frequency comb[23], chaotic optical communications[41,42], and chaotic laser imaging[19,43], opening up a variety of new possibilities for fundamental interest and practical applications. We anticipate that our findings will shed new light on the complex dynamics of nonlinear optical systems and further highlight the possibility for the development of new types of coherence-controlled chaotic laser sources.

## Methods

### Mode-locked fiber laser

A segment of 12-m erbium-doped fiber (EDF) with a dispersion parameter of $-17.3 \text{ ps km}^{-1} \text{ nm}^{-1}$ is employed as the gain medium. Other fibers used to construct the fiber laser are 18.05-m standard single-mode fiber (SMF). Thus, the fundamental repetition rate of the mode-locked fiber laser is 6.8 MHz. The polarization-insensitive isolator (PI-ISO) ensures unidirectional light propagation, while a polarization controller (PC) is employed to manipulate the polarization state of circulating light. The PC is used to slightly optimize the mode-locking state, which cannot switch the operation regime of the fiber laser at a fixed pump power. A saturable absorber (SA) is employed to achieve passive mode locking of the fiber laser. The flexible dispersion engineering is realized by a spectral pulse shaper (Finisar, Waveshaper 4000 A), where the imposed spectral phase profile $\phi$ that can be

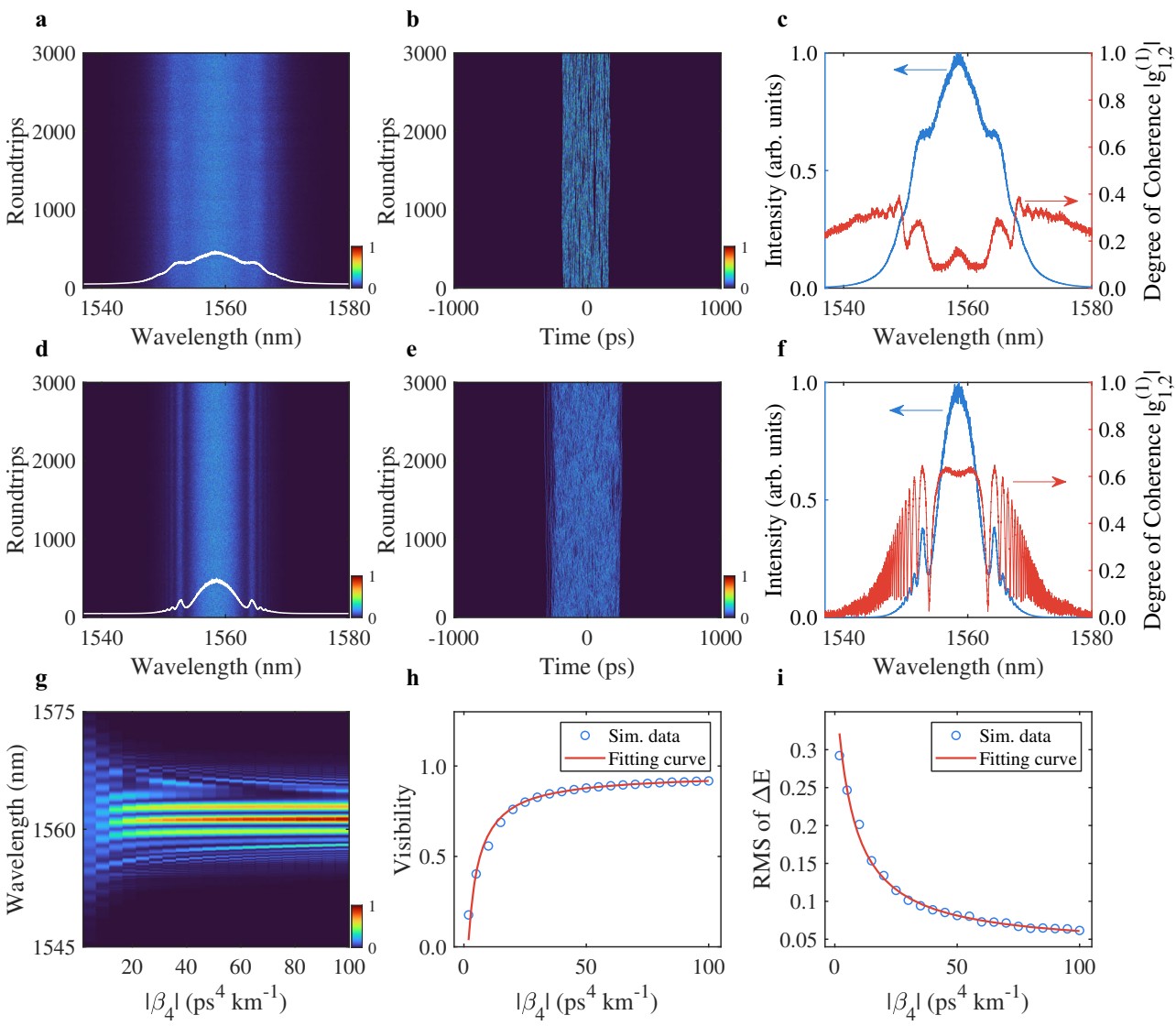

**Fig. 5 | Simulation results of coherence-controlled chaotic soliton bunch.**
**a–f** Characteristics of chaotic soliton bunch with (**a–c**) $\beta_4 = -1.9 \ \mathrm{ps^4 \ km^{-1}}$ (**d–f**) and $\beta_4 = -10 \ \mathrm{ps^4 \ km^{-1}}$, respectively. **a**, **d** Spectral and **b**, **e** temporal evolution. **c**, **f** Averaged spectrum (blue curve) and MDOC (red curve). **g–i** Evolutions of **g** spectral interference pattern, **h** visibility and **i** RMS of $\Delta E$ with increasing $|\beta_4|$.

expressed as:

$$\phi(\omega) = L \sum_{n=2}^{\infty} \frac{\beta_n \omega^n}{n!}, \qquad (1)$$

here, $L$ is the length of cavity, $\beta_n$ is $\mathrm{n^{th}}$-order dispersion coefficient and $\omega$ is the angular frequency. This implies that through the spectral pulse shaper, flexible dispersion control can be applied to the laser cavity. According to our experimental setup, we initially take $\beta_2 = +4.2642 \ \mathrm{ps^2 \ km^{-1}}$ for fully compensating GVD introduced by the optical fibers. Although there exists the FOD in optical fibers, the FOD in the conventional optical fiber is relatively small and can be generally ignored. Experimental characterization is performed on the output light beam extracted from the cavity by a 10:90 fiber output coupler (OC), which is simultaneously measured by an optical spectrum analyzer (OSA, Yokogawa AQ6375B) and a high-speed real-time oscilloscope (Tektronix DSA-70804, 8 GHz) with a photodetector (Newport 818-BB-35F, 12.5 GHz).

## DFT implementation

When the mode-locked pulse experiences a large GVD provided by the dispersive element, the spectrum of a pulse can be mapped to a temporal waveform whose intensity mimics its spectrum. This means that the spectrum can be captured by an oscilloscope with a fast photodetector in a real-time manner. Thus, DFT is a powerful method that overcomes the speed limitation of traditional spectrometers and hence enables fast real-time spectroscopic measurements. In our experiment, a ~15 km SMF with a dispersion parameter of 17 ps km$^{-1}$ nm$^{-1}$ is placed between the laser output and the photodetector, playing the role of the dispersive element with large GVD. Note that the spectral resolution of DFT is given by $\delta = \frac{1}{f|D|L}$, where $f$, $D$, and $L$ are the sampling rate of the oscilloscope, the dispersion parameter, and the length of the single-mode fiber. In this case, the spectral resolution of our DFT implementation is ~0.49 nm. When the pulse train comes out from the SMF, the spectrum of each pulse can be observed on the oscilloscope using the DFT technique. Therefore, the spectral evolution can be measured in a real-time diagnostic method.

## Phase coherence measurement of chaotic soliton bunch

The phase coherence of the chaotic soliton bunch is measured by injecting the solitons into an all-fiber Mach–Zehnder (M–Z) interferometer. The M–Z interferometer is composed of two 3-dB fiber couplers. The path difference between the two arms is set to be about one cavity roundtrip, which is precisely tuned by the fiber-compatible optical delay line on the shorter arm. The signal injected into the M–Z interferometer is then combined again by a 3-dB fiber coupler for interference. The spectral spacing of the interference fringe can also be adjusted by tuning the optical delay line. Note that the optical fiber that is used to construct the M–Z interferometer is fixed on the optical table for the purpose of stable operation.

## Numerical modeling

The numerical simulations allow us to better analyze the fiber laser system and explore the physical mechanism of the chaotic soliton bunch. The numerical simulations are based on the nonlinear Schrödinger equation, where the optical components are represented by the corresponding transfer functions:

$$\frac{\partial A}{\partial z} = -i\frac{\beta_2}{2}\frac{\partial^2 A}{\partial t^2} + i\frac{\beta_4}{24}\frac{\partial^4 A}{\partial t^4} + i\gamma|A|^2 A + \frac{g}{2}A + \frac{g}{2\Omega_g^2}\frac{\partial^2 A}{\partial t^2}, \quad (2)$$

here, $A$ is the slowly varying amplitude of the pulse envelope, and $i$ is the imaginary unit. The variables $z$ and $t$ represent the propagation coordinate and the time, respectively. $\beta_2$, $\beta_4$, $g$, and $\gamma$ are the GVD, FOD, gain coefficient, and Kerr nonlinearity of the fiber, respectively. $\Omega_g$ is the bandwidth of the gain spectrum. The gain coefficient of the EDF is represented by:

$$g = g_0 / \left(1 + E_p/E_s\right), \quad (3)$$

where $g_0$ is the small-signal gain, $E_p = \int |A(t)|^2 \, dt$ is the pulse energy, and $E_s$ is the saturable energy of EDF, which also represents the pump strength in practice. The transmission function of the amplitude representing the SA is:

$$T(t) = 1 - \frac{q_0}{1 + |A(t)|^2/P_0} - \alpha, \quad (4)$$

here, $q_0$ is the modulation depth of the SA, $|A(t)|^2$ is the instantaneous intensity, $\alpha$ is non-saturable loss and $P_0$ is the saturation power of the SA. The initial condition for numerical simulation is a sech-shaped pulse with relatively weak intensity in the time domain. As the pulse evolves in the cavity after one cavity round trip, the result is then used as the new initial signal in the next roundtrip for simulation. Moreover, to be consistent with our experiment, the following parameters are used for our simulations: $\Omega_g = 10$ nm, $g_0 = 0.8$ dBm$^{-1}$, and $\gamma = 0.003$ W$^{-1}$ m$^{-1}$, $\beta_2 = 22.3354$ ps$^2$ km$^{-1}$ and $L_{EDF} = 12$ m for the EDF; $\beta_2 = -21.9481$ ps$^2$ km$^{-1}$ and $L_{SMF} = 18.05$ m for the SMF. The modulation depth, non-saturable loss, and saturation power of SA are $q_0 = 0.21$, $\alpha = 0.14$ and $P_0 = 20$ W, respectively. The ratio of the laser output is $R_{out} = 10\%$. It is noted that we select the narrow effective gain bandwidth in simulations after considering the central wavelength of soliton and the working bandwidth of cavity elements in the experiment. Moreover, according to the experimental setup we used in this work, the initial $\beta_2$ is set to be $+4.2642$ ps$^2$ km$^{-1}$ in the spectral pulse shaper for compensating GVD introduced by the optical fibers. Then, $\beta_4$ is set to be different values for studying the dynamics of chaotic soliton bunch.

## Data availability

The data generated in this study have been deposited in the Zenodo database with access link of [https://zenodo.org/records/12508482].

## Code availability

The simulation codes of this study are available from the corresponding author upon request without any commercial interest.

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

## Acknowledgements

This work was supported by National Natural Science Foundation of China (62175069, 12274149, 62375091, 11874018); Basic and Applied Basic Research Foundation of Guangdong Province (2023B1515120044, 2023A1515011870); Guangzhou Science and Technology Plan Project (202201010202; 2024A04J5081).

## Author contributions

Z.L. and Z.Z. conceived the idea and designed the experiments. Z.Z. performed the experiments and conducted the numerical simulations with the help from Min L., J.L., Y.Y., T.L., A.L. and Z.L. Z.L., Meng L. and Z.Z. analyzed the results and prepared the manuscript. Z.L. and W.X. supervised the project. All the authors were closely involved in discussions and revised the manuscript.

## Competing interests

The authors declare no competing interests.
