## [Peer Review File · Nature Communications]

REVIEWER COMMENTS

Reviewer #1 (Remarks to the Author):

The manuscript by Zhang et al. proposed a new technology for controlling the coherence of chaotic soliton bunch in a mode-locked fiber laser. Basically, the topic is of great interest for nonlinear optics, and the idea is original. The authors explore the oscillating tails of optical solitons with fourth-order dispersion, and they come to the alleviation of the soliton fusion/annihilation from collisions in the chaotic bunch. Interestingly, they successfully demonstrate that the coherence of chaotic soliton bunch can be manipulated by managing fourth-order dispersion. In the meanwhile, the authors investigated the physical mechanisms behind achieving controllable coherence in the chaotic soliton bunch. It is noticed that there have indeed been previous studies on high-order dispersion for soliton shaping, the idea presented in this work, on the other hand, is novel and they successfully demonstrate the concept of "coherence-controlled chaotic soliton bunch." In particular, the authors present the application of such a coherence-controlled chaotic soliton bunch to manipulate the generation of optical rogue waves, an interesting concept for interdisciplinary studies. Complementary to the experimental studies, the numerical simulations additionally performed by the authors further verified their experimental results. Overall, I think the findings reported in this manuscript are exciting and provide a new route to realizing coherence control of a chaotic soliton bunch in a mode-locked laser, which is a high degree of complexity. I believe that this manuscript meets the criteria for publication in Nature Communications and I recommend acceptance after addressing the concerns provided below.

1. The oscillating tails of optical soliton result from the imperfect counterbalance between self-phase modulation - and FOD-induced phase shifts, which is critical to the coherence control of chaotic soliton bunch. I understand the origin of the oscillating tails of FOD-driven solitons (i.e., pure-quartic solitons). However, I believe it would be better for the readers if the authors could discuss the physics of the generation of oscillating tails in detail.

2. In the experimental section, the authors claimed that the fiber laser can enter the single soliton operation state or localized chaotic soliton bunch regime depending on the cavity parameters such as pump power. As there is a polarization controller in the laser cavity, I'm wondering whether other parameters, like states of polarization that can influence the operation regime of the mode-locked fiber laser.

3. For the demonstration of controlling the rogue waves, it is shown that the generation of optical rogue waves is suppressed when the introduced FOD is large enough, namely in the case of better coherence of chaotic soliton bunch. However, authors did not provide the criteria for how to identify the rogue waves.

4. For the visibility of the interference fringe, it seems that the visibility is not linearly changed with the increasing $\beta(4)$. Why? Please discuss this phenomenon with more details.

Some minor suggestions:

1. The authors are encouraged to explain more details about the “Delicate dispersion controlling”.
2. It would be better if the spectral range (wavelength) of DFT spectrum could align with those measured on the optical spectrum analyzer. In Figures 2c, 3c, and 3f, the spectral coordinate is not properly adjusted and needs corresponding corrections in the revised manuscript.
3. In Figures 3 and 5, the unit for x-axis is missed. It would be helpful to include the appropriate units.
4. "chaotic soliton bunch" (page 11) should be "chaotic soliton bunch". Please carefully proofread the manuscript during the revisions.

Reviewer #2 (Remarks to the Author):

The principle of controlling soliton pulse interaction through an adjustable amount of fourth-order dispersion has been theoretically proposed in 1994 (refs 26 and 30 of the manuscript). Therefore, experiments demonstrating such control in practice are highly welcome. The present manuscript attempts to demonstrate control of soliton interaction within a chaotic bunch of dissipative soliton pulses circulating round a fiber laser cavity. Chaotic soliton bunches in ultrafast lasers at large pumping power are indeed common phenomena. The manuscript shows interesting results of chaotic soliton bunches that feature a significant degree of coherence, which is evaluated from spectral interference between two consecutive cavity roundtrips, in presence of fourth-order dispersion (FOD).

However, the manuscript is unclear on many aspects, so that it is not possible to get any reliable conclusion, despite the strong assertions made in the conclusion section. I recommend an extensive revision, taking into account the following points:

1. The inclusion of FOD is made via a lumped element, a spectral pulse shaper. In addition, a laser is a dissipative system. Such situation departs from the initial concept of ref. 26, where the FOD is distributed in a passive fiber. Therefore, to make a possible connection, the authors should take a typical pulse duration and compare the dispersive length with respect to the cavity roundtrip length.
2. I don't find the two cases displayed in fig. 3 a-c and d-f so different, qualitatively, and even quantitatively. We would like to see a much larger variation of coherence to validate the idea of control. What is the coherence when the applied FOD is zero? The idea of control means that one should obtain a strong variation, and ideally a switch from low to high coherence with the

application of FOD beyond some threshold. This additional data, if conclusive, would make the report much more convincing.

3. In the experiment, the GVD is cancelled by the pulse shaper. Therefore, it is possible that the bunch coherence remains relatively high whatever the FOD, again dismissing the idea of a convincing control effect. I would strongly recommend the authors to start from a low coherence obtained with a certain amount of GVD, even small but not negligible, and then increase gradually the FOD to see an effect in coherence.

4. In figure 2, the “pulse train of the localized chaotic soliton bunch” is missing.

5. The numerical simulations should explore the proposal made in points 2 and 3 above.

6. In numerical simulations, the relaxation time of the saturable absorber is not included. This should be justified, comparing it to a typical pulse duration.

7. The introduction needs improvements and polishing. For instance, the following sentence “Thus, the chaotic soliton bunch delivered by GVD-dominated nonlinear system generally suffers from incoherent bunch-to-bunch phase” is very awkward. Indeed, by definition, one expects chaotic optical waveforms to be mostly incoherent! Also, note that oscillating pulse tails are not specific of solitons with FOD. For instance, dispersion-managed solitons and dissipative solitons are known to feature oscillating pulse tails.

The key point of the manuscript should be: can we adjust the amount of these oscillating pulse tails in a clear and quantitative way, in experiments, in order to control pulse interaction? This should be unambiguously demonstrated.

Point-by-Point Responses to Reviewers' Comments

We appreciate the reviewers very much for their positive and constructive comments that are quite helpful to improve the quality of our work. We have carefully digested the reviewers' comments and made appropriate revisions both in the main text and Supplementary Information, wherein the changes are highlighted in red. The detailed point-by-point responses to reviewers' comments are listed below.

Reviewer #1 (Remarks to the Author):

The manuscript by Zhang et al. proposed a new technology for controlling the coherence of chaotic soliton bunch in a mode-locked fiber laser. Basically, the topic is of great interest for nonlinear optics, and the idea is original. The authors explore the oscillating tails of optical solitons with fourth-order dispersion, and they come to the alleviation of the soliton fusion/annihilation from collisions in the chaotic bunch. Interestingly, they successfully demonstrate that the coherence of chaotic soliton bunch can be manipulated by managing fourth-order dispersion. In the meanwhile, the authors investigated the physical mechanisms behind achieving controllable coherence in the chaotic soliton bunch. It is noticed that there have indeed been previous studies on high-order dispersion for soliton shaping, the idea presented in this work, on the other hand, is novel and they successfully demonstrate the concept of "coherence-controlled chaotic soliton bunch." In particular, the authors present the application of such a coherence-controlled chaotic soliton bunch to manipulate the generation of optical rogue waves, an interesting concept for interdisciplinary studies. Complementary to the experimental studies, the numerical simulations additionally performed by the authors further verified their experimental results. Overall, I think the findings reported in this manuscript are exciting and provide a new route to realizing coherence control of a chaotic soliton bunch in a mode-locked laser, which is a high degree of complexity. I believe that this manuscript meets the

criteria for publication in Nature Communications and I recommend acceptance after addressing the concerns provided below.

Reply: We are thankful to the referee for his/her positive comments and suggestions. We have revised the manuscript accordingly. Listed below are our point-by-point responses to each of the referee's comments.

Comment 1: *The oscillating tails of optical soliton result from the imperfect counterbalance between self-phase modulation- and FOD-induced phase shifts, which is critical to the coherence control of chaotic soliton bunch. I understand the origin of the oscillating tails of FOD-driven solitons (i.e., pure-quartic solitons). However, I believe it would be better for the readers if the authors could discuss the physics of the generation of oscillating tails in detail.*

Reply: Thank the referee for the constructive comments. Indeed, as pointed out by the referee, the formation mechanism of oscillating pulse tails is not discussed in detail neither in the manuscript nor in the supplementary file. The origin of the oscillating pulse tails is only simply attributed to the imperfect counterbalance between self-phase modulation (SPM)- and FOD-induced phase shifts. In the following, we will discuss the formation mechanism of the oscillating tails of optical soliton that is balanced by SPM and FOD.

As it is well known to all, the SPM gives rise to an intensity-dependent phase shift. When the pulse propagating in the optical fiber is assumed as a Gaussian profile, the nonlinear phase shift is also temporally varying, which possesses a Gaussian profile, as depicted in Fig. R1 with blue dashed curve. As for the phase shift induced by the FOD, it has been demonstrated the phase shift profile has a quartic relationship with time, as plotted in Fig. R1 with red curve. Note that a critical condition for the stable propagation of a soliton in optical fiber is that the frequency chirp induced by SPM can be fully balanced by that of fiber dispersion. **However, from Fig. R1, it can be seen that only around the center part of the**

optical pulse can achieve a perfect balance between the SPM and FOD-induced phase shifts. For the two edges of the pulse, it is obvious that the phase shifts induced by SPM and FOD cannot be balanced. Therefore, if a Gaussian pulse is injected into an optical component with pure FOD, the energy of the optical pulse will flow from the center part to two edges owing to the imperfect counterbalance between SPM- and FOD-induced phase shifts. In this scenario, the oscillating tails can be observed with a soliton dominated by FOD and SPM.

Fig. R1. Phase profiles induced by SPM (blue curve) and FOD (red curve).

The formation of oscillating tails of the optical pulse leads to a variation of pulse intensity profile. As mentioned above, the SPM-induced phase shift is intensity-dependent. Thus, the SPM-induced phase shift will change according to the variation of the intensity pulse profile (that is to say, an optical pulse with oscillating tails). It has been demonstrated that the SPM-induced chirp is linear and positive (up-chirp) over a large central region of the optical pulse. In this case, the SPM-induced chirp would be positive across the main lobe of the optical pulse as well as the oscillating tails. On the other hand, the frequency chirp induced by FOD across the main lobe of the optical pulse and the oscillating tails is negative (down-chirp), as shown in Fig. R2. Therefore, this provides a condition for a balance between the SPM- and FOD-induced chirps, both for the main lobe of the pulse and oscillating tails. Combined with the optical soliton's self-organization effect, the optical soliton's stable propagation with the oscillating tails can be

achieved. This is the physical picture responsible for the formation of oscillating pulse tails in a FOD-dominated soliton. In order to more clearly show the formation mechanism of oscillating tails of FOD-driven pulse, we have also provided a detailed discussion in supplementary files. **Please see Supplementary Note 1.**

Fig. R2. Intensity profile of optical soliton with oscillating tails (blue dashed curve) and FOD-induced chirp profile (red curve).

Comment 2: *In the experimental section, the authors claimed that the fiber laser can enter the single soliton operation state or localized chaotic soliton bunch regime depending on the cavity parameters such as pump power. As there is a polarization controller in the laser cavity, I'm wondering whether other parameters, like states of polarization that can influence the operation regime of the mode-locked fiber laser.*

Reply: Indeed, we inserted a polarization controller into the ring cavity of the fiber laser. However, the mode-locking operation of the fiber laser is initiated by a saturable absorber in our experiment. The role of the polarization controller is to slightly optimize the mode-locking state relying on the weak polarization-dependent gain in the active fiber. In this sense, the operation regime of the mode-

locked fiber laser cannot be switched by only adjusting the polarization controller. For example, when the fiber laser is operating in the regime of chaotic soliton bunch, it will maintain this status even if the polarization controller is rotated in a large range. In fact, the operation regime of the passively mode-locked fiber laser is greatly influenced by the pump power, which actually determines the nonlinear phase shift experienced by the mode-locked soliton. With the overdriven nonlinear effect, namely by increasing the pump power, the single soliton will evolve into a chaotic soliton bunch, which is a typical feature of a soliton fiber laser. In order to more clearly define the role of the polarization controller, we have added a sentence in the revised manuscript, "The PC is used to slightly optimize the mode-locking state, which cannot switch the operation regime of the fiber laser at a fixed pump power."

***Comment 3:** For the demonstration of controlling the rogue waves, it is shown that the generation of optical rogue waves is suppressed when the introduced FOD is large enough, namely in the case of better coherence of chaotic soliton bunch. However, authors did not provide the criteria for how to identify the rogue waves.*

Reply: Indeed, we did not point out the criterion for identification of the optical rogue waves in the original manuscript, and we are sorry for missing it. As demonstrated in the manuscript, the generation of optical rogue waves can be effectively controlled by adjusting the amount of FOD that is introduced into the laser cavity. Here, we should stress that the general criterion of optical rogue wave generation is that the highest amplitude of waves is larger than twice the significant wave height (SWH) (*Phys. Rev. Lett.* **108**, 233901, 2012), and we also adopted this criterion for identifying the generation of optical rogue waves in our fiber laser. In order to directly point out the criterion for identifying the optical rogue waves, we have added a sentence, "Note that the criterion for identifying the optical rogue wave generation in this work is that the highest amplitude of waves is larger than twice the significant wave height (SWH), which is a widely adopted

criterion in investigating rogue waves according to previous reports⁵, in the Supplementary Note 9 of the revised version.

***Comment 4:** For the visibility of the interference fringe, it seems that the visibility is not linearly changed with the increasing beta(4). Why? Please discuss this phenomenon with more details.*

Reply: In our manuscript, the visibility of the interference fringe is introduced to indicate the coherence degree of the chaotic soliton bunch, which is closely linked to the intensity of the oscillating tails of the mode-locked pulse dominated by FOD. As demonstrated in our work, the intensities of the oscillating tails are determined by the amount of the imposed FOD in the laser cavity. That is to say, the coherence of chaotic soliton bunch (or the visibility of the interference fringe) is closely related to the role of FOD on pulse shaping. As is well known before, **the role of group velocity dispersion on pulse shaping can be intuitively reflected by the quadratic dispersion length**, which is defined by $L_{GVD} = T_0^2 / |\beta_2|$ (T_0 is pulse width and β_2 is GVD). Similar to that of GVD, the role of FOD on pulse shaping can also be qualitatively reflected by the quartic dispersion length, which is defined as $L_{FOD} = T_0^4 / |\beta_4|$ (*Nat. commun.* **7**, 10427, 2016). In this case, **the effect of FOD on the pulse shaping (for example, the intensity of oscillating tails) is inversely proportional to the FOD amount**. That is to say, the visibility of the interference fringe or the coherence of chaotic soliton bunch does not linearly change with the increasing FOD. According to the referee's comments, we have added the following sentences in the revised manuscript to discuss this phenomenon more clearly, “Here, we can see that the fringe visibility does not linearly change with the increasing FOD. It is because the effect of FOD on pulse shaping can be qualitatively reflected by the dispersion length, which is defined as \$L_{FOD} = T_0^4 / |\beta_4|\$. This leads to a nonlinear relationship between the fringe visibility and the FOD amount. Moreover, although a lumped FOD engineering is adopted, the calculated dispersion length (see Supplementary Note 5) indicates that the effect of FOD on

pulse shaping is still evident during the process of coherence-controlled operation when compared to the laser cavity length.”

Comment 5: The authors are encouraged to explain more details about the “Delicate dispersion controlling”.

Reply: The dispersion control is realized by a commercial programmable spectral pulse shaper in our experiments. In fact, **the spectral pulse shaper can impose different phase masks on the injected optical pulse by virtue of a spatial light modulator (SLM), which is flexible and accurate for dispersion control.** The basic operation principle is as follows: firstly, a grating is used to disperse the spectral components of the input light onto the pixel array plane of the SLM. Then, by externally applying an electric field to control the refractive index of each liquid crystal pixel unit, the phase of different frequency components is modulated, achieving arbitrary shaping of the input light spectrum. Since the dispersion effect is manifested as a phase response in the frequency domain, arbitrary-order dispersion control can be realized through a programmable spectral pulse shaper. Compared to the conventional dispersion management approach, i.e., changing the length of dispersive optical fiber, the use of a spectral pulse shaper can easily introduce a desirable dispersion management. That is why we label it as delicate dispersion controlling in Fig. 1.

Comment 6: It would be better if the spectral range (wavelength) of DFT spectrum could align with those measured on the optical spectrum analyzer. In Figures 2c, 3c, and 3f, the spectral coordinate is not properly adjusted and needs corresponding corrections in the revised manuscript.

Reply: Thank the referee for the kind reminder. Indeed, in the original submission, we did not consider the alignment of spectral coordinates for the spectra recorded by the DFT and the conventional optical spectrum analyzer. According to the

referee's suggestion, we have adjusted the spectral coordinates of the figures, which are all related to the spectra measured by DFT and the optical spectrum analyzer. Please see the Figures in the revised manuscript.

Comment 7: *In Figures 3 and 5, the unit for x-axis is missed. It would be helpful to include the appropriate units.*

Reply: We appreciate the referee very much for pointing it out. According to the comment, we have added the unit for the x-axis in Figures 3 and 5. Please see the revised version.

Comment 8: *"chaotic solion bunch" (page 11) should be "chaotic soliton bunch". Please carefully proofread the manuscript during the revisions.*

Reply: We are very sorry for making such a typo in the manuscript. Now, we have corrected it in the revised version. In addition, we have proofread the whole manuscript carefully and made the necessary corrections regarding the grammar or typos. Please see the revised manuscript.

Reviewer #2 (Remarks to the Author):

The principle of controlling soliton pulse interaction through an adjustable amount of fourth-order dispersion has been theoretically proposed in 1994 (refs 26 and 30 of the manuscript). Therefore, experiments demonstrating such control in practice are highly welcome. The present manuscript attempts to demonstrate control of soliton interaction within a chaotic bunch of dissipative soliton pulses circulating round a fiber laser cavity. Chaotic soliton bunches in ultrafast lasers at large pumping power are indeed common phenomena. The manuscript shows interesting results of chaotic soliton bunches that feature a significant degree of coherence, which is evaluated from spectral interference between two consecutive cavity roundtrips, in presence of

forth-order dispersion (FOD).

However, the manuscript is unclear on many aspects, so that it is not possible to get any reliable conclusion, despite the strong assertions made in the conclusion section. I recommend an extensive revision, taking into account the following points:

Reply: Thank the referee very much for the critical comments, which are quite helpful in improving the quality of the manuscript. We made great efforts to address all comments that were raised by the referee. Detailed point-by-point responses are given below.

Comment 1: *The inclusion of FOD is made via a lumped element, a spectral pulse shaper. In addition, a laser is a dissipative system. Such situation departs from the initial concept of ref. 26, where the FOD is distributed in a passive fiber. Therefore, to make a possible connection, the authors should take a typical pulse duration and compare the dispersive length with respect to the cavity roundtrip length.*

Reply: We would like to thank the referee for the thoughtful comment. Indeed, the initial concept for the soliton interactions with oscillating tails was investigated numerically by distributing the FOD in a passive optical fiber. However, for the experiment, as far as we know, there is no such optical fiber that holds the features of large and tunable FOD operation. In this case, the lumped element (i.e., programmable spectral pulse shaper) is generally introduced to impose the FOD on pulse shaping in a laser cavity. Note that the lumped FOD has been used to shape the mode-locked soliton recently, where the optical soliton with obvious oscillating tails is generated by employing the lumped FOD effect, namely pure-quartic soliton (*Nat. Photonics* **14**, 492 2020). **This work actually demonstrated that the lumped way for FOD introduction is feasible and effective, where FOD plays a dominated role in pulse shaping in a laser cavity.** In addition, with the development of fiber fabrication technology, photonic crystal fiber may be a promising candidate for possessing pure FOD as well as the FOD tunable operation

(*Opt. Express* **26**, 7786, 2018). In this way, we are able to build a fiber laser in which pure FOD is distributed along the fiber cavity.

As indicated by the referee, the dispersion length is a good criterion for checking the role of FOD on mode-locked soliton shaping in a laser cavity. For our laser setup, the cavity length is ~ 30.05 m. According to the autocorrelation trace, the averaged pulse duration of the solitons inside the chaotic bunch is about 1 ps. Note that the quartic dispersion length is defined as $L_{\text{FOD}} = T_0^4 / |\beta_4|$. In the meanwhile, we have found that the pulse durations of the chaotic solitons do not change much during the process of FOD adjustment. Therefore, the quartic dispersion lengths with the FOD variations can be calculated, as plotted in Fig. R3. As can be seen here, the L_{FOD} is reaching 1000 m when the FOD is set to be $\beta_4 = -1 \text{ ps}^4 \text{ km}^{-1}$. Considering the cavity length is 30.05 m, the effect of FOD on pulse shaping seems to be weak. **However, we should note that, in order to enhance the effect of FOD on pulse shaping, the GVD is compensated to be 0 in our experiment.** In this case, it can be observed that the coherence of the chaotic soliton bunch is still influenced by FOD-driven pulse shaping, even if the FOD is not large. In particular, when the FOD is increased to $-10 \text{ ps}^4 \text{ km}^{-1}$, which corresponds to a quartic dispersion length of 100 m, the coherence-controlling effect of the chaotic soliton bunch becomes evident. This point has also been verified in our experiment (see Fig. 3 of the revised version). Then, the effect of FOD on pulse shaping is strong when the FOD is increased to $\beta_4 = -30 \text{ ps}^4 \text{ km}^{-1}$, that is, the dispersion length associated with FOD is comparable to the cavity length. The evolution curve of dispersion length with the FOD also further verifies that the coherence of chaotic soliton bunch relies on the role of FOD, as the evolution curves of the quartic dispersion length and the coherence of chaotic soliton bunch with an increasing FOD exhibit the same trends. Therefore, according to the dispersion lengths associated with different FODs and laser cavity length, it can be concluded that the pulse shaping is still dominated by FOD even if the lumped way of dispersion compensation is adopted. In order to more clearly

describe the role of lumped FOD on pulse shaping, we have added a more detailed description in the revised manuscript: “Here, we can see that the fringe visibility does not linearly change with the increasing FOD. It is because the effect of FOD on pulse shaping can be qualitatively reflected by the dispersion length, which is defined as \$L_{\text{FOD}} = T_0^4 / |\beta_4|\$. This leads to a nonlinear relationship between the fringe visibility and the FOD amount. Moreover, although a lumped FOD engineering is adopted, the calculated dispersion length (see Supplementary Note 5) indicates that the effect of FOD on pulse shaping is still evident during the process of coherence-controlled operation when compared to the laser cavity length.”

Fig. R3. Quartic dispersion length with varying $|\beta_4|$.

Comment 2: I don't find the two cases displayed in fig. 3 a-c and d-f so different, qualitatively, and even quantitatively. We would like to see a much larger variation of coherence to validate the idea of control. What is the coherence when the applied FOD is zero? The idea of control means that one should obtain a strong variation, and ideally a switch from low to high coherence with the application of FOD beyond some threshold. This additional data, if conclusive, would make the report much more convincing.

Reply: We appreciate the referee for the constructive comments, which definitely is a critical issue that needs to be improved. Indeed, in the original version of the manuscript, the two cases regarding the coherence-controlled operation presented in Fig. 3 look similar to each other. It is because the tuning range of FOD to control the coherence of chaotic soliton bunch is not large enough, which cannot fully manifest itself as the ideal of “coherence control”. According to the referee’s comment, we have redone the experiment on the coherence control of chaotic soliton bunch with a much larger range of FOD variation. **Correspondingly, the visibility of the spectral interference fringe, which is used to evaluate the coherence of chaotic soliton bunch, can be controlled from low to high, i.e., from 0.27 to 0.71 in the experiment and 0.18 to 0.92 in the numerical simulation, respectively,** as presented in Fig. R4 below. The difference in the coherence variation range between the experiment and the simulation lies in the detection sensitivity and noise in the experiments. Here, it should be noted that the fiber laser cannot maintain the regime of the chaotic soliton bunch when the FOD is adjusted to 0. This is because the operation regime of a passively mode-locked fiber laser is largely related to the balance between the dispersion and nonlinearity. However, when FOD is introduced in the laser (even if the FOD is small), for example, $\beta_4 = -1 \text{ ps}^4\text{km}^{-1}$ in the laser cavity, the fiber laser is operating in the regime of chaotic soliton bunch, and a low modulation depth of the interference fringe can be observed on the mode-locked spectrum after passing the M-Z interferometer, as shown in Fig. R5a. This case was also verified in the numerical simulations, which is well consistent with the experimental observations, see Fig. R5b. **With these additional data, we now think the idea of coherence control has been demonstrated with much more solid evidence.** According to the referee’s comments, we have re-plotted the dataset regarding the coherence-controlled operation with new ones and re-written the corresponding descriptions. Please see Figure 3 of the revised manuscript.

Fig. R4. Coherence of chaotic soliton bunch varies with $|\beta_4|$. **a** experimental result, **b** simulation result.

Fig. R5. Interference fringe of chaotic soliton bunch. **a** experimental result, **b** simulation result.

Comment 3: *In the experiment, the GVD is cancelled by the pulse shaper. Therefore, it is possible that the bunch coherence remains relatively high whatever the FOD, again dismissing the idea of a convincing control effect. I would strongly recommend the authors to start from a low coherence obtained with a certain amount of GVD, even small but not negligible, and then increase gradually the FOD to see an effect in coherence.*

Reply: We fully agree with the referee and thank the referee for the thoughtful comments. Initially, **we conducted both the experiments and simulations for the coherence control of chaotic soliton bunch under the condition of GVD**

cancellation by the spectral pulse shaper because we wanted to enhance the effect of FOD on pulse shaping in our laser cavity. Indeed, investigating the coherence property of chaotic soliton bunch with a certain amount of GVD and a varying FOD would lead to a more general conclusion made in this work, which would definitely help improve the manuscript quality. As demonstrated above, the coherence of the chaotic soliton bunch is increasing nonlinearly with the increasing FOD. According to the referee's suggestion, we have investigated the coherence property of chaotic soliton bunch with a certain amount of GVD ($\beta_2 = -4.2624 \text{ ps}^2 \text{ km}^{-1}$ here) and a varying FOD in both experiment and numerical simulation. **It was found that the conclusion of coherence control of chaotic soliton bunch is still well established with FOD variation.** Here, the visibility of the interference fringe is adjusted from 0.54 to 0.73 in experiment when the FOD is increasing from $-5 \text{ ps}^4 \text{ km}^{-1}$ to $-100 \text{ ps}^4 \text{ km}^{-1}$, showing an evident trend from low to high coherence, as plotted in Fig. R6. That is to say, a certain amount of GVD introduced in the laser cavity does not break the conclusion of coherence control of the chaotic soliton bunch by adjusting the FOD. Nevertheless, owing to the existence of GVD, the achievable controlling range of the bunch coherence is smaller than the case of GVD cancellation, because the pulse dynamics of the fiber laser is dominated by GVD when the FOD is initially small. At this point, the effect of GVD will impact the mode-locking state. This is also why the GVD is compensated to be 0 to investigate the coherence-controlled property of chaotic soliton bunch in the original submission. **In addition, we have also added the results of coherence evolution with a certain amount of GVD, as well as the corresponding descriptions in the Supplementary Note 6.**

Fig. R6. Characteristics of chaotic soliton bunch with a certain amount of GVD and a varying FOD. a experimental and **b** simulation results of conference evolution of chaotic soliton bunches, **c** experimental and **d** simulation results of RMS of intensity fluctuation of chaotic soliton bunch.

Comment 4: In figure 2, the “pulse train of the localized chaotic soliton bunch” is missing.

Reply: Thank the referee for the careful comment. We are sorry that we have made such a mistake in the original submission. In fact, we initially intended to provide the pulse train of the chaotic soliton bunch as an inset of Fig. 2. But later, we found that the pulse train did not contain much useful information except for the pulse repetition rate and intensity uniformity. Unfortunately, we forgot to delete the figure caption for the original submission. Because there is no pulse train provided in the whole manuscript as well as the supplementary material, after careful

consideration, we decided to provide the pulse train of the localized chaotic soliton bunch in the revised manuscript, as shown in Fig. R7 below, and please also see the inset of Fig. 2b of the revised manuscript. Moreover, we have added a sentence to describe it, "The pulse train is also plotted (inset of Figure 2b), which shows an evident intensity fluctuation owing to the partially coherent property."

Fig. R7. Pulse train of localized chaotic soliton bunch.

Comment 5: *The numerical simulations should explore the proposal made in points 2 and 3 above.*

Reply: According to the referee's suggestion, we have carried out the numerical simulations regarding the coherence property of chaotic soliton bunch with a larger range of FOD variation and a certain amount of GVD. The results can be referred to Reply to comments 2 and 3. It can be seen that the coherence evolutions of the numerical simulations are consistent with those of the experimental ones. Again, we would like to thank the referee for the helpful comments.

Comment 6: *In numerical simulations, the relaxation time of the saturable absorber is not included. This should be justified, comparing it to a typical pulse duration.*

Reply: We appreciate the referee for making a professional comment on our simulation model. Indeed, the relaxation time of the saturable absorber is not

considered in the simulation model of our fiber laser, which might play an important role in describing pulse shaping in some situations. For example, when the interaction between the optical pulse and the saturable absorber needs to be described precisely, the relaxation time of a saturable absorber should be considered. In addition, when the pulse duration delivered by a laser is much shorter than the relaxation time of the saturable absorber, the relaxation time also might be considered, as in this case a large distortion of the pulse profile can be observed during the shaping process.

However, the fiber laser generally delivers a pulse train with a duration ranging from hundreds of femtoseconds to several picoseconds, where the saturable absorber used in the experiment can be regarded as a fast saturable absorber taking into account its relaxation time. This is also the case for our work, while the typical pulse duration is about 1ps. In fact, **most of the pulse dynamics in a fiber laser can be implemented by a simplified model of a saturable absorber in the numerical simulations.** That is, in the numerical simulations, we categorize the saturable absorber as a fast one. The model of a fast saturable absorber without consideration of the relaxation time has been widely adopted to simulate the pulse dynamics in fiber laser (*IEEE J. Sel. Top. Quantum Electron.* **4**, 159, 1998; *Nat. Photonics* **14**, 492, 2020), which can be described as:

$$q = 1 - \frac{q_0}{1 + |A(t)|^2/P_0} \quad (1)$$

where P_0 is the saturation power of the saturable absorber and q_0 is the modulation depth. Here, it is supposed that the effect of nonlinear transmission q acts instantaneously on the pulse intensity $P(t) = |A(t)|^2$. In addition, the simulation results without considering the saturable absorber's relaxation time can reproduce the experimental phenomena well in our work. Therefore, we did not include the relaxation time of the saturable absorber in our simulation model, and we thank the referee again for the thoughtful comment.

Comment 7: *The introduction needs improvements and polishing. For instance, the*

following sentence “Thus, the chaotic soliton bunch delivered by GVD - dominated nonlinear system generally suffers from incoherent bunch - to - bunch phase” is very awkward. Indeed, by definition, one expects chaotic optical waveforms to be mostly incoherent! Also, note that oscillating pulse tails are not specific of solitons with FOD. For instance, dispersion-managed solitons and dissipative solitons are known to feature oscillating pulse tails.

Reply: Thank the referee very much for the meticulous comments. Indeed, inaccurate English usage leads to a misunderstanding of the meaning of what we actually want to express. As suggested by the referee, we have rewritten the sentence “Unfortunately, owing to a lack of dynamic equilibrium mechanism for soliton collisions within the bunch, chaotic solitons formed in these systems exhibits randomly evolving phases and amplitudes among solitons, causing numerous and stochastic fusions/annihilations during collision process. Thus, the chaotic soliton bunch delivered by GVD-dominated nonlinear system generally suffers from incoherent bunch-to-bunch phase.” to be “In fact, the chaotic solitons formed in these systems exhibit randomly evolving phases and amplitudes, causing numerous and stochastic fusions/annihilations during the collision process. Owing to a lack of a dynamic equilibrium mechanism for stochastic soliton interactions within the bunch, it remains a challenging task to control the coherence of a chaotic soliton bunch²⁴.” in the revised manuscript.

As for the generation of oscillating pulse tails, indeed, it is not a unique feature of solitons with FOD, and it can also be observed in other types of solitons. According to the referee’s comment, we have also revised the description of the oscillating tails of the mode-locked soliton dominated by FOD, “Due to an imperfect counterbalance between self-phase modulation (SPM)- and FOD-induced phase shifts, the oscillating tails appear on both edges of the soliton^{28,29}. From a general perspective of soliton interactions, the oscillating tails of FOD-driven soliton would play a vitally important role in soliton interactions³⁰.” **Corresponding, we have provided a more detailed discussion on the physical**

mechanism responsible for the formation of oscillating tails of the FOD-driven soliton in the supplementary files. Please see Supplementary Note 1 of the revised version. In addition, we have also polished the whole manuscript to make the expression more logical and precise. Please see the revised manuscript.

***Comment 8:** The key point of the manuscript should be: can we adjust the amount of these oscillating pulse tails in a clear and quantitative way, in experiments, in order to control pulse interaction? This should be unambiguously demonstrated.*

Reply: We appreciated the referee's constructive comments. Indeed, the key to controlling the coherence of a chaotic soliton bunch is to adjust the intensity of oscillating pulse tails, which play an important role in the soliton interactions within the bunch that avoid the soliton fusions/annihilations during the collision process. We are sorry that we did not provide the intensity evolution of the oscillating pulse tails by varying the FOD value in the original submission. To demonstrate such an adjustment of the oscillating tails, we employed a home-made ultrafast fiber laser to investigate the intensity evolution of the oscillating pulse tails. The home-made ultrafast laser source is then passed through a spectral pulse shaper, where the chirp of the mode-locked pulse induced by GVD is compensated to be near 0, and the FOD can be flexibly adjusted by virtue of the programmable spectral pulse shaper. The pulse profile of the final output is recorded by a commercial autocorrelator. The experimental setup is shown in Fig. R8 below.

Fig. R8. Setup for the control of the oscillating tails.

By changing the FOD amount experienced by the optical soliton, it is clearly observed that the intensity of the oscillating pulse tails can be flexibly adjusted, as shown in Fig. R9. Here, it should be noted that owing to the limitation of the

autocorrelator, only a pedestal can be observed on the autocorrelation trace when the FOD amount imposed on the mode-locked soliton is larger, i.e., -2.4 ps^4 . It can be observed that, as the introduced amount of FOD increases gradually, the pedestal of the autocorrelation trace becomes larger. **The gradually strengthening pedestal in the autocorrelation trace reflects the increasing intensity of the oscillating pulse tails.** Hence, we think these results are enough to demonstrate the adjustment of oscillating pulse tails, and therefore to control the soliton interactions. To further validate the experiment and display more details, a numerical simulation of the experimental process was also conducted, with the results shown in Fig. R10. The simulation results show that, as the amount of FOD increases, the temporal envelope exhibits larger oscillating tails, which is well consistent with the experimental findings. In order to discuss this issue more clearly, we have added the experimental and simulation results regarding the adjustment of oscillating pulse tails through FOD variation, as well as the corresponding descriptions in the supplementary files. **Please see the Supplementary Note 2 in the revised version.**

Fig. R9. Experimental results of the autocorrelation trace under the action of different FOD

amounts. **a** 0 ps^4 , **b** -0.8 ps^4 , **c** -1.6 ps^4 , **d** -2.4 ps^4 .

Fig. R10. Simulation results of the temporal pulse envelope under the condition of different FOD amounts. a 0 ps^4 , **b** -0.8 ps^4 , **c** -1.6 ps^4 , **d** -2.4 ps^4 .

REVIEWER COMMENTS

Reviewer #2 (Remarks to the Author):

I have carefully read the authors' responses as well as the revised manuscript. I think the authors have provided a detailed responses to all my concerns. Particularly, they have performed extra experiments and numerical simulations to provide more dynamics of coherence-controlled chaotic soliton bunch, and they presented efficient coherence control of the chaotic soliton bunch under larger coherence variation. Now I have no further comment, and recommend it for publishing in Nature Communications.

Reviewer #3 (Remarks to the Author):

The authors have seriously considered my criticism and recommendations and provided additional evidence and explanations, which make me feel more comfortable with a possible recommendation for publication in Nature Communications. For the sake of physical understanding and clarity, I would nevertheless request the authors to provide additional information about the following points.

1- A key point that still needs to be improved is the correct consideration about lumped and distributed effects. Actually, there has been some confusion around my previous comment #1. What I mean is that, for a single pulse (constituent of the pulse bunch), of expected duration range 0.5ps-1ps, such pulse experiences the dominant GVD dispersive effects of the optical fibers, which are 12-m (EDF, normal D) and 18-m (SMF, anomalous D) long. It seems that the dispersion lengths (GVD) should be around the length of these fibers (10-20 meters). This may be close to a limiting situation where the dynamic effects of a local GVD cannot just be compensated/avoided. However, this limiting case could still work to highlight the influence of FOD, as suggested by the authors. The authors could make a clearer statement around these physical facts.

2- Then, we have to figure out what are typical FOD the values – or their order of magnitude – in these EDF and SMF fibers. Maybe it is easier to calculate a typical FOD for a standard SMF-28 fiber. Have the authors done so? Otherwise, can we measure the total FOD in the experiment by fitting the spectral sidebands? We would like to compare the effects of distributed FOD (in fibers) and lumped (from pulse shaper) to make sure about which magnitude of added FOD really controls the overall dynamics: is it $1 \text{ ps}^4/\text{km}$ or $10 \text{ ps}^4/\text{km}$?

3- The author response to my comment 6 is not very convincing, as they use a commercial semiconductor SA, whose relaxation time seems to be 2 ps, namely not negligible compared to the pulse duration. Therefore, it could also have some impact in pulse-pulse interaction, when two pulses get close, separated by less than 2 ps: this would make an effective repulsive force that would help improving the bunch coherence, akin to the FOD! However, I agree that it is still reasonable, in a first attempt, to simulate as the authors did, without taking the SA relaxation time. Including the SA relaxation time in the model should be a next interesting task in subsequent investigations.

4- Considering my comment 8, it would be much more convincing to operate the laser at a lower pump power to generate one or a few pulses (namely, not in the chaotic bunch case) and study the evolution of the autocorrelation trace according to the level of FOD implemented by the wave shaper, to check that indeed the pulse tails increase with the applied FOD. Can this be done experimentally?

Point-by-Point Responses to Reviewers' Comments

Reviewer #2:

I have carefully read the authors' responses as well as the revised manuscript. I think the authors have provided a detailed responses to all my concerns. Particularly, they have performed extra experiments and numerical simulations to provide more dynamics of coherence-controlled chaotic soliton bunch, and they presented efficient coherence control of the chaotic soliton bunch under larger coherence variation. Now I have no further comment, and recommend it for publishing in Nature Communications.

Reply: We appreciate the reviewer's positive comments and insightful suggestions for improving our manuscript during the first round of review.

Reviewer #3:

The authors have seriously considered my criticism and recommendations and provided additional evidence and explanations, which make me feel more comfortable with a possible recommendation for publication in Nature Communications. For the sake of physical understanding and clarity, I would nevertheless request the authors to provide additional information about the following points.

Reply: We thank the reviewer for the constructive and insightful comments on improving our manuscript in the first round of review. According to the reviewer's additional in-depth comments, we provide the point-to-point responses as follows.

Comment 1: *A key point that still needs to be improved is the correct consideration*

about lumped and distributed effects. Actually, there has been some confusion around my previous comment #1. What I mean is that, for a single pulse (constituent of the pulse bunch), of expected duration range 0.5ps-1ps, such pulse experiences the dominant GVD dispersive effects of the optical fibers, which are 12-m (EDF, normal D) and 18-m (SMF, anomalous D) long. It seems that the dispersion lengths (GVD) should be around the length of these fibers (10-20 meters). This may be close to a limiting situation where the dynamic effects of a local GVD cannot just be compensated/avoided. However, this limiting case could still work to highlight the influence of FOD, as suggested by the authors. The authors could make a clearer statement around these physical facts.

Reply: We appreciate the referee for the in-depth comments that are quite constructive for our further studies on the dynamics of chaotic soliton bunch dominated by FOD. As indicated by the referee, the local dynamics of the mode-locked pulses can still be influenced by the distributed GVD along the SMF and EDF, although the cavity GVD is compensated by the spectral pulse shaper. Indeed, the dispersion lengths associated with GVD in the SMF and EDF are comparable to their individual physical length in our laser cavity. And this will make the local dispersive effect caused by GVD non-negligible either in the SMF or EDF. In this scenario, the mode-locked soliton as well as the oscillating tails will be broadened in the EDF and compressed in the SMF dynamically in the laser cavity, owing to the local distributed GVD along the SMF and EDF. However, it has been demonstrated that the output characteristics of a mode-locked fiber laser are predominantly determined by the net cavity dispersion. That is to say, although the mode-locked soliton as well as the oscillating tails experiences broadening or compression along the propagation either in the SMF or EDF, the major features of the mode-locked pulse are still dependent on the net cavity dispersion.

In our experiment, the cavity GVD is compensated to be 0 in a lumped way by the spectral pulse shaper, so as to reduce the impact of the GVD greatly on the intra-cavity pulse shaping. In this case, the effect of distributed GVD along the SMF

and EDF on the broadening and compression of the mode-locked soliton and the oscillating tails can be also compensated in the spectral pulse shaper. Therefore, FOD is able to play a dominant role in pulse shaping when FOD is introduced into the laser cavity by the spectral pulse shaper. Benefitting from your suggestions during the first round of review, we have also experimentally and numerically investigated the coherence-controlled property of the chaotic soliton bunch with a small amount of cavity GVD. **Although the residual cavity GVD will weaken the role of FOD on the coherence-controlled ability to some extent, the conclusion of coherence control of chaotic soliton bunch is still well established.** This result demonstrated that the local dispersive effect of the optical fibers caused by GVD could influence the coherence-controlled operation of the chaotic soliton bunch, if the cavity GVD is not compensated properly.

Here, we would like to thank the referee again for the thoughtful comments, as we believe that the effect of distributed GVD along the fibers on the coherence control of a chaotic soliton bunch would be meaningful for our further investigations. According to the referee's comments, we have rewritten a sentence to make a clearer statement on this issue, "Moreover, we have also checked the coherence property of chaotic soliton bunch with a certain amount of GVD and a varying FOD. Although the residual distributed GVD along the optical fibers will weaken the role of FOD on the pulse shaping, the coherence-controlled operation is still well established."

***Comment 2:** Then, we have to figure out what are typical FOD the values – or their order of magnitude – in these EDF and SMF fibers. Maybe it is easier to calculate a typical FOD for a standard SMF-28 fiber. Have the authors done so? Otherwise, can we measure the total FOD in the experiment by fitting the spectral sidebands? We would like to compare the effects of distributed FOD (in fibers) and lumped (from pulse shaper) to make sure about which magnitude of added FOD really controls the overall dynamics: is it $1 \text{ ps}^4/\text{km}$ or $10 \text{ ps}^4/\text{km}$?*

Reply: Thank the referee for the meticulous comments. Indeed, we did not discuss the FOD of SMF and EDF in our manuscript. But actually, the FOD in the standard SMF is relatively small, which can be generally ignored. For example, in the work by Runge et al. (Nat. Photonics 14, 492 2020), the FOD of the optical fiber they adopted is about $-0.0022\text{ps}^4/\text{km}$. In this case, although the FOD that is introduced by the spectral pulse shaper is $-1\text{ps}^4/\text{km}$, the compensated FOD amount is about 500 times larger than that of optical fiber.

On the other hand, it is true that FOD exists in optical fibers. Nevertheless, the conventional chaotic soliton bunch in a fiber laser without additional introduction of FOD shows a relatively low bunch-to-bunch coherence and an uncontrollable coherence. **This means that the variation of FOD amount from the spectral pulse shaper is essential to control the coherence of the chaotic soliton bunch.** Indeed, the coherence of the chaotic soliton bunch can be flexibly adjusted by applying different amount of FOD into the laser cavity, as demonstrated in our work. According to the referee's suggestion, we have also calculated the total FOD of our fiber laser by fitting the spectral sidebands when the different FOD amounts are applied from the spectral pulse shaper [$\beta_4 = -10\text{ps}^4/\text{km}$ (blue), $\beta_4 = -20\text{ps}^4/\text{km}$ (green), $\beta_4 = -100\text{ps}^4/\text{km}$ (purple)]. The results are shown in Fig. R1 below. Then the cavity FOD is calculated to be $\beta_4 = -9.9963\text{ps}^4/\text{km}$ (blue), $\beta_4 = -20.3295\text{ps}^4/\text{km}$ (green), $\beta_4 = -100.2259\text{ps}^4/\text{km}$ (purple). So, we can conclude that cavity FOD is critically dependent on the lumped FOD introduced by spectral pulse shaper, and the overall dynamics of the fiber laser is really controlled by the lumped FOD. According to the referee's comment, we have added a sentence in the "methods" section, "Although there exists the FOD in optical fibers, the FOD in the conventional optical fiber is relatively small and can be generally ignored."

Fig. R1. Cavity FOD calculation according to the positions of spectral sidebands. a Averaged spectra. **b** The fourth power of the sideband positions plotted against the sideband order. From top to bottom, $\beta_4 = -10$ ps⁴/km (blue), $\beta_4 = -20$ ps⁴/km (green), $\beta_4 = -100$ ps⁴/km (purple).

***Comment 3:** The author response to my comment 6 is not very convincing, as they use a commercial semiconductor SA, whose relaxation time seems to be 2 ps, namely not negligible compared to the pulse duration. Therefore, it could also have some impact in pulse-pulse interaction, when two pulses get close, separated by less than 2 ps: this would make an effective repulsive force that would help improving the bunch coherence, akin to the FOD! However, I agree that it is still reasonable, in a first attempt, to simulate as the authors did, without taking the SA relaxation time. Including the SA relaxation time in the model should be a next interesting task in subsequent investigations.*

Reply: We appreciated the referee very much for the constructive comments. Indeed, when the relaxation time of a SA is around 2 ps, the impact of relaxation time of a SA on the pulse-pulse interaction through an effective repulsive force might improve the coherence of a chaotic soliton bunch to some extent in the fiber laser. **However, this property cannot lead to a control operation of the bunch-to-bunch coherence, as the repulsive force generated by SA can be considered as a fixed force from an averaging point of view.** Moreover, as discussed in our work, the coherence of a conventional chaotic soliton bunch

would be relatively low when purely relying on the repulsive force among the solitons generated by SA. These results, in turn, demonstrated the critical role of the FOD on the coherence-controlled operation of a chaotic soliton bunch. Anyway, we would like to thank the referee again for the thoughtful comments, because the consideration of the SA relaxation time in the simulation model will definitely give rise to more interesting nonlinear phenomena, and we hope to figure it out in the near future.

***Comment 4:** Considering my comment 8, it would be much more convincing to operate the laser at a lower pump power to generate one or a few pulses (namely, not in the chaotic bunch case) and study the evolution of the autocorrelation trace according to the level of FOD implemented by the wave shaper, to check that indeed the pulse tails increase with the applied FOD. Can this be done experimentally?*

Reply: We fully agree with the referee. Indeed, it would be more convincing if we can measure the autocorrelation trace evolution of a stable pulse directly from the fiber laser with different amounts of FOD. In fact, we have also tried to do so during the first round of review. However, it fails because the pump power should be set to a relatively low pump power to generate one or a few stable pulses. Note that the intensity of the oscillating tails is relatively low when comparing to the main lobe of the mode-locked soliton. Moreover, the sensitivity of the used autocorrelator is not high. In this case, we cannot measure the evident oscillating tails in our experiment.

On the other hand, we provide the simulation results on the evolution of temporal profiles of a stable pulse directly from the fiber laser with different amounts of FOD. The results are presented in Fig. R2 below, in which we can see that intensity of the oscillating tails is stronger with the increasing FOD. In addition, in the first round of review, we have also demonstrated the oscillating tails can be effectively generated if the mode-locked soliton is passed through a FOD component, namely spectral pulse shaper. Therefore, it can be concluded that the

intensity of the oscillating tails is indeed increasing with the increasing FOD.

Fig. R2. Simulation results of the temporal pulse envelope under the condition of different FOD amounts. **a** -0.03 ps^4 , **b** -0.3 ps^4 , **c** -1.2 ps^4 , **d** -3.01 ps^4 .